# Cereal Commodity Trading in Ethiopian Local Marketplace: Examining Farmers' Quantity Measurement Behaviors

**Kidane Assefa Abebe [1]**, **Deyi Zhou [1,\*]**, **Bekele Gebisa Etea [1]**, **Fekadu Megersa Senbeta [2]**, **Dereje Kebebew Debeli [3]** and **Rajani Osti [1]**

[1] College of Economics and Management, Huazhong Agricultural University, No. 1 Shizishan Street, Hongshan District, Wuhan 430070, China; kidane2016@yahoo.com (K.A.A.); bgebisa@yahoo.com (B.G.E.); ostirajani@yahoo.com (R.O.)

[2] School of Economics, College of Resource and Environmental Economics, Huazhong University of Science and Technology, Luoyu Road 1037, Wuhan 430074, China; fekadu.megersa.fm@gmail.com

[3] Key Laboratory of Textile Science and Technology, Ministry of Education, College of Textiles, Donghua University, Shanghai 201620, China; drjdebeli@yahoo.com

\* Correspondence: zdy@mail.hzau.edu.cn; Tel.: +86-155-2782-6338

**Abstract:** Local marketplaces are remarkable organizations for agricultural product transactions in Ethiopia. However, little is known concerning measurement practices in these micro-trading zones. Thus, this study intended to examine the cereal commodity quantity measurement behaviors of farmers in the local marketplaces of Ethiopia. A survey was conducted in four districts marketplaces ($N = 382$) of the Oromia Region. The $\chi^2$ test was employed to evaluate the association between farmers' perception of the accuracy of local units and measuring instruments related to immoral buyers' behaviors. According to the findings, farmers' cereals quantity measurement behaviors proved the presence of unreliability which created transaction, measurement, social capital, and two-hand palm cereals gift costs. The $\chi^2$ test results indicated that farmers' perceptions of the accuracy of local units and measuring tools related to buyers' unethical behaviors had significant relationships with bowl, glass, sack, and can local units, except for weight balance in Dendi and Bako Tibe, and for cans in the Gimbichu area. This study demonstrates that standardization of tools and measurements, together with institutional support, would have a huge potential for economizing transaction costs and making equitable cereals exchanges and efficient markets.

**Keywords:** economic transaction; measurement behaviors; local marketplace; local units

## 1. Introduction

The concept of measurement is associated with different economic concerns and decisions. These concerns and decisions entail what products to produce and how to produce them (measurement of specifications), what quantities of products are traded and how this estimate is reliable (measurement of quantity), and what the condition the products are in and how functional they are (measurement of quality) [1,2]. These economic measurement structures are the basic elements and incentives of economic transactions [3,4]. Institutional economics literature also considers the importance of measurement costs, which is part of total transaction costs [5]. Aoki [6] gives emphasis on well-functioning market institutions (in this context the local agricultural marketplace) within which people practice measurements. In this regard, homogenous measurement units and measurement systems have a significant contribution to manage costly transfer of resources which occur due to unreliable and non-uniform measures [1]. Besides, the consistency in measurement units notably

resolves complicated internal trade, are local in nature, and enhance the integration of markets in a given nation [7,8].

Scholars identified standardized measures and institutions (rules of governing trade) to determine the economic transaction measurement problems that inhibit economic agents from attaining maximum gains [7,9–13]. Nevertheless, many developing countries such as Ethiopia have not had access to Imperial and Metric measurement systems for a long period of time because the systems remained in the hands of the inventors [14]. Even after the adoption of standardized measures, the target of many nations was on metrology standardization mainly by the state administration [9]. In most developing countries, the multiple and non-uniform local units are still widely in use though governments have implemented the metric system [14–16]. In addition, it has not been adequately investigated whether metrological standardization *per se* solves entire transactional problems or not [1,2]. Moreover, contemporary economics literature proposes institutional structures for managing measurement issues of economic trade [12,17]. In another way, institutions that are supposed to govern market participants' unethical behaviors can create measurement costs. But the basic issue that seeks scientific investigation is how to formulate workable institutions that will better perform in local marketplaces. Additionally, there is no evolution of market institutions' frameworks that can be a threshold for recent local measurement systems [18,19].

The origin of measurement problems have so far been investigated from different perspectives. According to Barzel [4], measurement problems arise from measurements errors, which compare the actual measurement to the standard. This view holds a premise that the management of standardized measures can ensure a measurements' reliability. On the other hand, the Velker's approach of measurement sameness considers the source of measurement problems broadly from reliability dimensions [2]. This approach believes that standardized measures are used only to assist the management of measurements. Herein, we can obviously understand that the metrological standardization roles are varied in scope in ensuring measurement reliability. In general, there is no apparent fact that indicates which approach can fully address the whole source of measurement problems, particularly in local agricultural marketplace trading environments.

In Ethiopian rural marketplaces, farmers trade agricultural products using multiple, non-uniform, and incoherent local units of measurement. These conditions lead trading parties to be involved in acts of abuse, injustice, and violence, which in turn create uncertainties and costs. Most of these problems emerge from bulged sizes of local units, using the large size of measuring instruments apart from pre-agreed units, count deception, and measuring below the standard of measurement units. In fact, the multiplicity of measuring tools and system impact is not only regarded at the micro-level, but also disturbs the macro-economy. A reliable dataset of commodities marketed are fundamental to make a sound economic analysis and design macro-level policy. But the accuracy level of locally-measured information concerning the quantities of traded and consumed products found using heterogeneous units and measurements is still indistinguishable to construct rigorous macro-economic policies. Conversely, scholars have little conception of how measurements are currently being practiced in micro-trading zone contexts [1]. Besides, policy-makers have paid less attention towards farmers' measurements costs, which involves agricultural products traded in light of its consequences on the local economy.

In light of the above concerns and scholarly arguments, this study is aimed at examining cereal quantity measurement behaviors to address local agricultural marketplace measurement reliability, exchange equitability, and institutional development. In the present study, farmers' trading partners' measuring instruments related to immoral norms of behaviors were also investigated. In addition, the study analyzed the diverse measurement approaches and assumptions based on the responses from real contexts of local market organizations. Overall, the study paves a way to generate economic knowledge and further assists the management of measurement costs.

## 2. Materials and Methods

### 2.1. Description of Study Area

Oromia Regional State is an autonomous region in Ethiopia. The aggregate cereal production of this region comprised about 44.5 percent of the country in 2015 [20]. East and West Shoa zones were chosen among the six popular zones in cereal production in the region [21] (Figure 1). Dendi and Bako Tibe districts are chosen in the West Shoa Zone. These districts are located between the geographic coordinates of 8°45′–9°10′ N and 37°56′–38°20′ E and 8°56′– 9°13′ N and 37°00′–37°17′ E, respectively. The Dendi and Bako Tibe districts have an average annual temperature of 9.3–23.3 °C and 9–25 °C, and average rainfall of 900–1300 mm and 900–1281 mm, respectively. Agro-ecological zones of vast areas of Dendi and Bako Tibe are stratified into high land and mid high land.

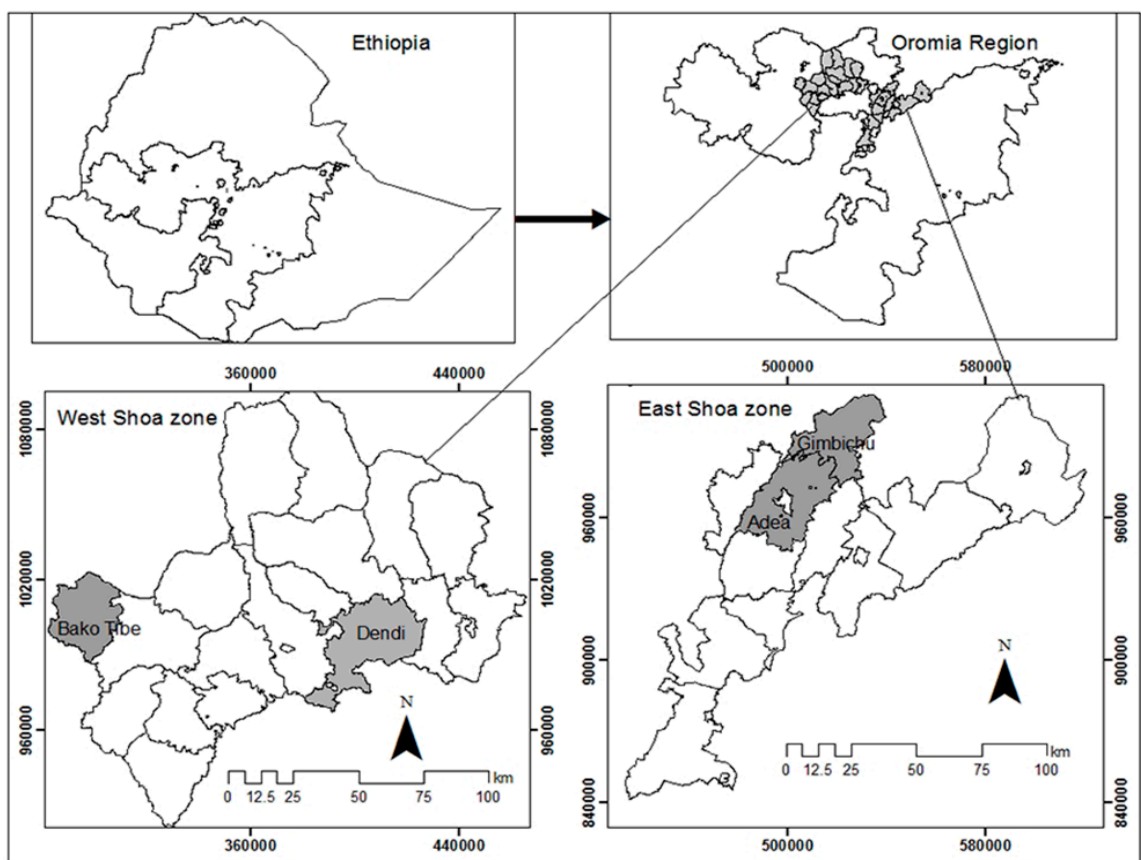

**Figure 1.** Overview of the study area.

The Adea and Gimbichu districts are located in the East Shoa Zone. The districts are located between 8°22′–8°56′ N and 38°58′–39°22′ E and 8°49′–9°09′ N and 38°58′–39°22′ E, respectively. They have an average annual temperature of 17.6–25 °C and 15–22.5 °C, respectively. The vast area of both districts get an average rainfall ranging from 900 to 1300 mm. The districts are stratified into two agro-ecological zones based on agro-climatic conditions namely: dega 5 and 95 percent and weindega 95 and 40 percent in Adea and Gimbichu, respectively.

The 2007 national census report reveals that a total population of 165,803; 123,031; 130,321; and 86,902 reside in Dendi, Bako-tibe, Adea, and Gimbichu districts, respectively. The selected areas are popular in producing cereals. Based on the data obtained from the zonal administrations, the total post-harvest cereals production of East and West Shoa zones in 2015 was 7,965,315 and 15,652,419 quintals (1 quintal is equivalent to 100 kg), respectively [22]. Of which, 42.9 and 14.31 percent were produced in Adea and Gimbichu (East Shoa) and Dendi and Bako Tibe (West Shoa), respectively.

## 2.2. Sampling Method and Sample Size

The study areas were selected using purposive and random sampling methods. The selection was based on the relative volume of cereal production [20] and proximity to the capital city of the country. On top of this, the cereal commodity consumption expenditures of the household [23] and the cereal trade center factor were considered in the study area selection. To this end, four district (Bako Tibe, Dendi, Adea, and Gimbichu) marketplaces were identified for the study. Of these chosen areas, sample sizes for the farmer respondents were determined by using the formula [24] denoted in Equation (1) with the assumption of a 5 percent level of precision; 95 percent level of confidence; and 50 percent degree of variability.

$$n_o = \frac{z^2(p)(q)}{e^2} \tag{1}$$

where $n_o$ is the sample size, $z$ is the abscissa of the normal curve (the value of $z$ is taken from the statistical tables of the normal curve), $e$ is the desired level of precision, $p$ is the estimated proportion of an attribute that is present in the farmer population, and $q$ is $1 - p$. Since the total population is finite and known, the sample size ($n_o$) was adjusted using Equation (2).

$$n = \frac{n_o}{1 + \frac{n_o - 1}{N}} \tag{2}$$

Accordingly, the total sample size ($N$ = 382) was divided into the four districts' marketplaces depending on their respective number of farmer households. Hence, the sample sizes were 91, 124, 90, and 77, Bako Tibe, Dendi, Adea, and Gimbichu marketplaces, respectively.

## 2.3. Data Types and Methods of Collection

This study employed both primary and secondary data. The primary data were gathered using the survey method through administered structured questionnaires from 1 March to 30 May 2017. The survey was mainly focused on the farmers' socio-economic characteristics, cereal commodity measurement trends, farmers' perceptions about local measurement units, and buyers' measurement related behaviors. Moreover, local marketplace observation was conducted to supplement the survey. Secondary data was obtained from office reports, journals, books, and working papers.

## 2.4. Method of Data Analysis

In this study, data was analyzed using the 23rd version of the Statistical Package for Social Sciences (SPSS) (IBM corporation, New York, NY USA), specifically through descriptive statistics such as percentage, mean, and the chi-square test. The OriginPro 9.1 (OriginLab Corporation, Guangzhou, China) data analysis and graphing software was also employed to indicate farmers' age, education, and cereal commodity trading experience in the marketplace.

## 3. Results and Discussion

The survey results revealed that there were more percentages of male farmers (66.9, 61.5, 89.6, and 50) in the Dendi, Bako, Gimbichu, and Adea marketplaces, respectively. The percentage of farmers engaged in marriage was 71.8, 72.5, 64.9, and 73.3 in the Dendi, Bako, Gimbichu, and Adea marketplaces, respectively. As displayed in Figure 2a–c, sampled farmers' age, education, and cereals trade experience distribution was almost comparable through all the study areas except in the Bako marketplace.

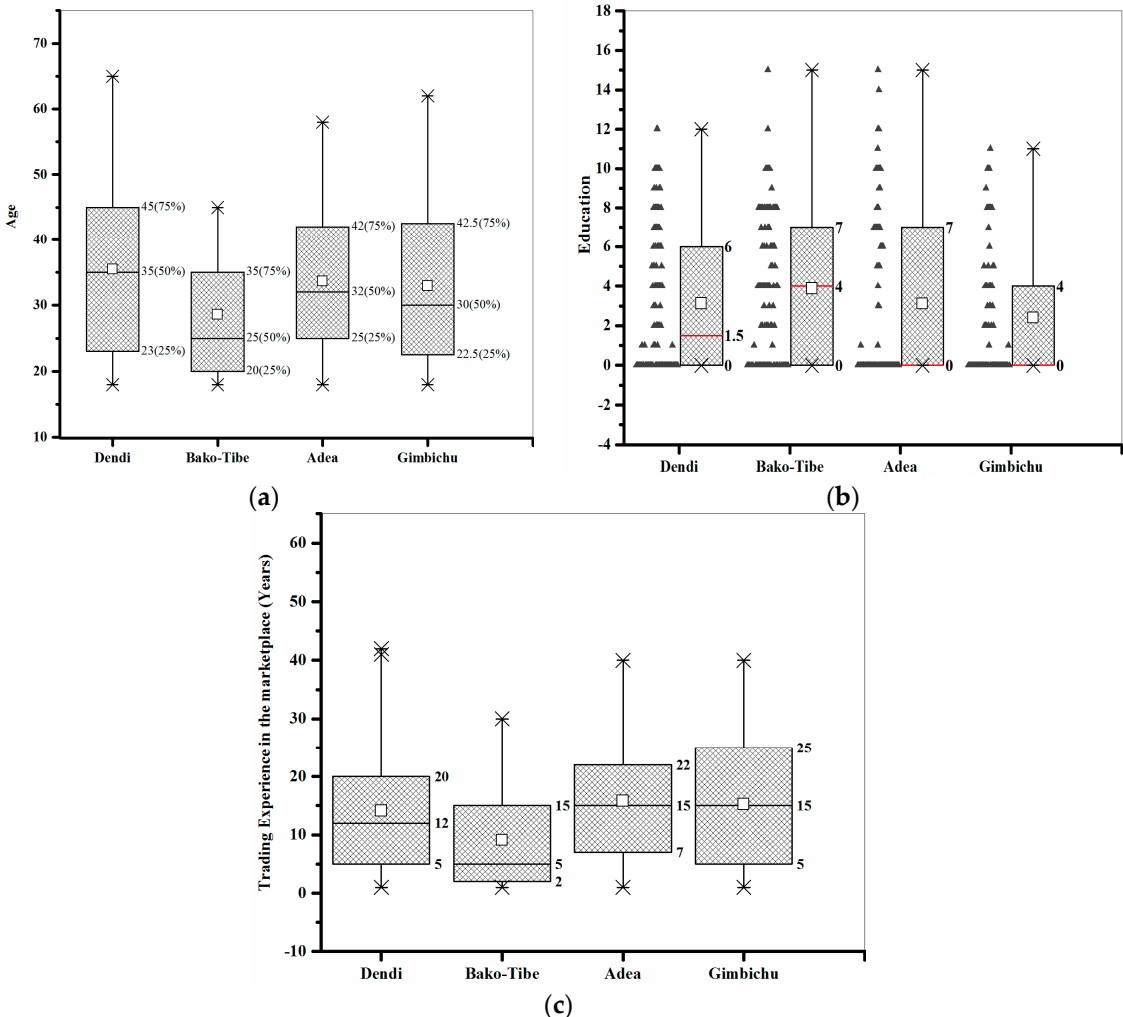

**Figure 2.** (**a**) Age; (**b**) education; and (**c**) cereal commodity trading experience of the farmers.

*3.1. Quantity Measurement Behaviors of Cereal Commodity*

3.1.1. Local Measures and Conversion Convention

The findings of this study exhibited that multiple and non-uniform measuring instruments were applied for cereals trading (Table 1a). In particular, 75 percent of the samples in the Dendi marketplace were using bowl (locally named Kelelle) instruments. In contrast, glass was a major volumetric local unit in Bako district, as exposed by 85.7 percent of the samples. Mechanical weight balance was used for a few crops transactions under certain conditions. The market asserted that cereals grown on clay soil have a high weight compared to similar types of crops grown upon other types of soils. Based on this market agreement, 25 percent of the sample respondents were using mechanical balance typically for selling teff cereal, which grows on clay soil in Dendi District (Table 1a). However, the weight differences among crops harvested on different types of soil requires further exploration. On the other hand, in the Bako Tibe site, maize crops were dominantly exchanged using mechanical balance (Table 1a). Farmers were additionally selling teff and wheat using balance when the market price was comparatively better than local glass units (Table 1b).

In Adea and Gimbichu, the cereal quantity measurement units used in the marketplace were quite unique. In these locations, two sub-market centers were located side-by-side depending on the kind of local units. Most of Gimbichu's farmers were exchanging macro-cereal commodities using an empty sack of urea fertilizer, which has been issued each year since 2009 (Table 1a). Farmers were using a full

sack of commodity by sewing it with lean string as a unit of measurement. On the contrary, two types of sacks—an empty sack of dap and of urea fertilizer—were used simultaneously and interchangeably for the large quantity of cereal trade in the Adea marketplace (Table 1a). The methods of using a sack unit in Adea is different from that of the Gimbichu area. Farmers were using a sack to measure a crops quantity. They usually fill up the sack to the maximum edge without sewing it. Here, the lack of a common pre-agreed way of using a sack instrument probably causes quantity variation between the two areas. Moreover, the remaining samples were mostly using a can unit branded White Oats, Bebelac, or Infacare for micro-trade in Gimbichu and Adea (Table 1a).

The study, thus, concluded that each district executes its own independent local units of measurement. Two to three general kinds of local measuring units were also employed in each marketplace based on the kind and quantity of commodity in trade (Table 1a). Beyond these, non-uniform local measures were vastly seen even for the same kind of local units such as the sack, can, glass, and bowl. Other kinds of units with similar names were also applied for cereals transactions. In addition, the study revealed that marketplace conversion conventions of local units' mass, that were equivalent to a kilogram (metric unit), were different among farmers of each district. For example, 72.6 percent of Dendi sample farmers agreed that two bowls of cereals is equivalent to three kilograms. Two bowl masses of cereals were tantamount to two kilograms for the rest of the samples (Table 1a). In addition, most farmers (68.1 percent) from the Bako marketplace believed that three glasses of cereal quantity are equal to one kilogram (Table 1a). In some research sites, sample participants were estimating local units' cereal amount to kilogram considering the issued year of the unit of measurement, for instance, the sacks used for local units (Table 1a). For other local units, like the cans, 12.2 percent of farmers in Adea agreed on the similarity between 1 and $\frac{1}{4}$ cans of cereals quantity and one kilogram. Whereas, 1 and half cans of cereal quantity is equal to one kilogram for the majority of farmers in Gimbichu's marketplace (Table 1a). The concern here is that the local instruments' cereals amounts of similar type were not identical, and thus confirmed the opportunity for costly transfer of cereals quantity.

The aforementioned findings, therefore, clearly demonstrated the unreliability of measurements, which led to non-equitable cereals exchanges. Under the situations of this case study, comparing the cereals trade economy among the districts' farmers was another immense problem. Theoretically, the presence of multiple and non-uniform local measures create uncertainties and costs, make markets non-modern and complicating internal trade [4,25,26]. The disparity of units and conversion conventions of on-hand cases similarly signified the existence of measurement costs. In such cases, complete standardization of cereals quantity measurement systems and institutions (market rules) are jointly essential to reduce local marketplace measurement problems.

### 3.1.2. Local Units' Ownership, Selection, and Negotiation Conflicts

In Ethiopia, local marketplace parties (i.e., the farmer and buyer) use their own measuring units. Thus, it is crucial to investigate ownership and trade parties that decide the selection of appropriate measuring units. Accordingly, the study showed that most units of measurement applied for trading belonged to buyers in Dendi (53.2 percent) and Bako (76.9 percent) areas (Table 1a). In contrast, farmers' measuring instruments were largely used in the Adea and Gimbichu study areas (Table 1a). The decision to use a suitable measures from the alternatives was made by farmers in most study sites (Table 1a). In contrast, the majority of buyers (52.7 percent) had more bargaining power to pass the decision over to local units in Bako (Table 1a). In some contexts, both farmers and buyers were jointly deciding and selecting an appropriate unit (Table 1a). Analytically, local measures ownership issues and the decision to select units of measurement were complicated.

**Table 1.** Cereals quantity measurement behavior of farmers.

| Items | | Research Site | | | |
|---|---|---|---|---|---|
| | | Dendi (Count %) | Bako (Count %) | Adea (Count %) | Gimbichu (Count %) |
| **(a)** | | | | | |
| Farmers' measuring unit for trading cereals | Mechanical Balance | 31 (25.0) | 13 (14.3) | 0 (0) | 0 (0) |
| | Can | 0 (0) | 0 (0) | 18 (20) | 11 (14.3) |
| | Glass | 0 (0) | 78 (85.7) | 0 (0) | 0 (0) |
| | Bowl | 93 (75) | 0 (0) | 0 (0) | 0 (0) |
| | Sack | 0 (0) | 0 (0) | 72 (80) | 66 (85.7) |
| Number of local units used within the marketplace | 2 kinds | 124 (100) | 91 (100) | 49 (54.4) | 77 (100) |
| | 3 kinds | 0 (0) | 0 (0) | 41 (45.6) | 0 (0) |
| Marketplace conversion convention of local unit mass that equivalent to a kilogram | 1.25 Cans | 0 (0) | 0 (0) | 11 (12.2) | 3 (3.9) |
| | 1.5 Cans | 0 (00 | 0 (0) | 7 (7.8) | 8 (10.4) |
| | 2 Bowls = 2 kg | 34 (27.4) | 0 (0) | 0 (0) | 0 (0) |
| | 2 Bowls = 3 kg | 90 (72.6) | 0 (0) | 0 (0) | 0 (0) |
| | 3 glasses | 0 (0) | 62 (68.1) | 0 (0) | 0 (0) |
| | 4 glasses | 0 (0) | 29 (31.9) | 0 (0) | 0 (0) |
| | Depending on the issued year of local unit | 0 (0) | 0 (0) | 72 (80) | 66 (85.7) |
| Ownership of local unit applied for trade | Farmer | 38 (30.6) | 15 (16.5) | 79 (87.8) | 68 (88.3) |
| | Buyer | 66 (53.2) | 70 (76.9) | 11 (12.2) | 5 (6.5) |
| | Marketplace neighbors | 20 (16.1) | 6 (6.6) | 0 (0) | 4 (5.2) |
| Trading parties deciding upon local unit that apply to trade cereals | Farmer | 44 (35.5) | 25 (27.5) | 82 (91.1) | 74 (96.1) |
| | Buyer | 43 (34.7) | 48 (52.7) | 7 (7.8) | 0 (0) |
| | Both parties | 37 (29.8) | 18 (19.8) | 1 (1.1) | 3 (3.9) |
| Negotiation conflicts related to instrument selection (farmers' experiences) | Yes | 65 (52.4) | 58 (63.7) | 33 (36.7) | 15 (19.5) |
| | No | 59 (47.6) | 33 (36.3) | 57 (63.3) | 62 (80.5) |
| Actor responsible for measuring cereal quantity | Farmer | 93 (75) | 78 (85.7) | 90 (100) | 77 (100) |
| | Buyer | 31 (25) | 13 (14.3) | 0 (0) | 0 (0) |
| Total (*n*) | | 124 (100) | 91 (100) | 90 (100) | 77 (100) |

**Table 1.** *Cont.*

| Items | | Dendi (Count %) | Bako (Count %) | Adea (Count %) | Gimbichu (Count %) |
|---|---|---|---|---|---|
| | | | **Research Site** | | |
| | | **(b)** | | | |
| Farmers' home-based quantity measuring habit | Yes | 27 (21.8) | 62 (68.1) | 2 (2.2) | 1 (1.3) |
| | No | 97 (78.2) | 29 (31.9) | 88 (97.8) | 76 (98.7) |
| The difference between home and marketplace measurements | Yes | 27 (21.8) | 62 (68.1) | 0 (0) | 1 (1.3) |
| | No | 0 (0) | 0 (0) | 2 (2.2) | 0 (0) |
| | Don't know | 97 (78.2) | 29 (31.9) | 88 (97.8) | 76 (98.7) |
| Quantity difference between home (HM) and marketplace (MPM) quantity measurements | HM volume is greater than MPM | 23 (18.5) | 58 (63.7) | 0 (0) | 1 (1.3) |
| | MPM volume is greater than HM | 4 (3.2) | 4 (4.4) | 0 (0) | 0 (0) |
| | Both are equal | 0 (0) | 0 (0) | 2 (2.2) | 0 (0) |
| | Don't know | 97 (78.2) | 29 (31.9) | 88 (97.8) | 76 (98.7) |
| Average quantity variance per sack | HM are greater than MPM | 2.4 bowls | 11.8 glasses | - | - |
| Average quantity variance per sack | MPM is greater than HM | 1 bowl | 5 glasses | - | - |
| The culture of offering commodity gift | Yes | 93 (75) | 78 (85.7) | 0 (0) | 0 (0) |
| | No | 31 (25) | 13 (14.3) | 90 (100) | 77 (100) |
| Average number of buyers a farmer transacts exchanges with | Farmers' experience | 2.7 | 2.5 | 1.4 | 1.4 |
| Number of times the cereal gift delivered to a buyer | None | 31 (25) | 13 (14.3) | 90 (100) | 77 (100) |
| | 1 time | 93 (75) | 77 (84.6) | 0 (0) | 0 (0) |
| | 2 times | 0 (0) | 1 (1.1) | 0 (0) | 0 (0) |
| The extent of relationship farmer has with buyers | No relationship at all | 82 (66.1) | 70 (76.9) | 69 (76.7) | 69 (89.6) |
| | Strong relationship | 42 (33.9) | 21 (23.1) | 21 (23.3) | 8 (10.4) |
| Why mechanical balance is not vastly used | Reducing trade benefits | 69 (55.6) | 65 (71.4) | 49 (54.4) | 57 (74) |
| | Farmers are unable to read and use a balance | 7 (5.6) | 0 (0) | 2 (2.2) | 0 (0) |
| | Using balance is not the culture of the marketplace | 4 (3.2) | 4 (4.4) | 12 (13.3) | 8 (10.4) |
| | Balance is used for teff and wheat when price is better than using other local units | 0 (0) | 3 (3.3) | 3 (3.3) | 0 (0) |
| | Farmers are using balance for maize exchange | 0 (0) | 13 (14.3) | 0 (0) | 0 (0) |
| | Difficult to find and use balance in the marketplace | 0 (0) | 1 (1.1) | 13 (14.4) | 0 (0) |
| | Farmers are using balance for teff, pulses and oilseeds | 33 (26.6) | 0 (0) | 0 (0) | 0 (0) |
| | Buyers are not willing to buy if balance is in use | 0 (0) | 0 (0) | 7 (7.8) | 8 (10.4) |
| | Farmers are afraid of cheating | 11 (8.9) | 5 (5.5) | 4 (4.4) | 4 (5.2) |
| Total (*N*) | | 124 (100) | 91 (100) | 90 (100) | 77 (100) |

Source: Field survey, 2017.

However, negotiation conflicts related to measuring instrument selection were relatively higher when buyers' ownership of measures and decisions were dominant. For instance, 52.4 and 63.7 percent of sample farmers from Dendi and Bako supported this assertion, even though farmers were accountable for measuring trade items (Table 1a). This phenomenon is, in fact, creating social capital costs for the parties in general. Such social capital costs, in turn, have the probability of increasing the total duration of time spent on conducting transactions, i.e., transaction costs. In another way, it is obvious that the exchange ties among actors are most pertinent to the economization of transaction costs. In this regard, a new institutional economics theory suggests trading aid in light of a homogenous unit of measurement application [27]. Some economists have offered further ways of managing measurement that include improved governance of transactions, third-party monitoring, and guarantees [1].

### 3.2. Home-Based Cereals Measuring Norms and the Two-Hand Palm Gift

The study analyzed farmers' norms of measuring cereals quantity before transporting to the marketplace. Thus, sample farmers in two districts have norms of measuring cereals at their home (Table 1b). The sample farmers who have such norms identified the variation between home and marketplace quantity measurements (Table 1b). Specifically, 63.7 and 18.5 percent of Bako and Dendi respondents, respectively, have these measuring norms. These respondents recognized that the home-based quantity measurement was greater than that of the marketplace. The average quantity difference per one sack was 11.8 glasses and 2.4 bowls in Bako and Dendi marketplace, respectively. However, few respondents replied that the quantity measurement at the marketplace was higher than home on average by 1 bowl and 5 glasses in Dendi and Bako, respectively. The findings generally showed different measurements of cereals amount at these two places. Comparatively, the home-based measurements mass were higher than the marketplace. In another assertion, the results proved the presence of measurement costs either by the non-uniformity of local units or dissimilar methods of measuring using a particular instrument.

Moreover, the norms of offering cereals gifts for the buyer was another part of the local marketplace measurement system. According to this study, cereals quantity gift behavior was performed within two districts (Table 1b). Consequently, 75 and 85.7 percent of sample farmer participants were offering a cereal gift for each buyer in the Dendi and Bako sites, respectively. In these trading sites, farmer respondents sold their total cereal supply to 2.6 buyers on average (Table 1b). The most samples were offering a two-hand palm gift only one time for a buyer (Table 1b). Over total commodity supply, each farmer had the possibility to deliver the gift for more than two buyers. However, the extent of the relationship between the farmers and buyers was almost inconsequential for most of the farmers across the sites. In this regard, the two-hand palm gift offered for two buyers on average can be logically considered as costs. Because it was not universal across district marketplaces and local units, and the relationship between farmers and buyers (expected outcome of the gift) was almost weak (Table 1b). For instance, 66.1, 76.9, 76.7, and 89.6 percent of farmers have no relationship with buyers in Dendi, Bako, Adea, and Gimbichu district, respectively.

The study farther indicated that the farmer respondents did not provide cereal commodity gifts when mechanical balance was in use. However, farmers were not using balance assuming that the balance reduces the trade benefit; they are unable to read the amount and use it, and using balance is not culturally recognized and difficult to find and use (Table 1b). The existence of cheating by distorting balance was also used (Table 1b). In some sites, farmers were using balance for few crops (Table 1b). In general, mechanical balance was not adequate to solve cereals transactional problems. Hence, the supposition of Barzel [4] that claimed the correspondence of measures and measurement is, therefore, not supported by such local marketplace realities. In the context of this study, mechanical balance is only helping to make cereal quantity measurements. Therefore, establishing institutions are essential to solve such measurement problems.

*3.3. Farmers' Perception of Buyers' Measurements-Related Unethical Behavior and Local Units' Accuracy*

Informal norms of behaviors and social conventions are important to govern actors' behaviors [28]. Social norms and conventions provide a structure for an individual and lessen uncertainty within everyday life. Such norms and conventions are immensely practical to run economic transactions in an open market structure. More specifically, this study intended to investigate the farmers' perception of buyers' cereals quantity measurements related immoral norms of behaviors.

The findings revealed that the buyers enormously practiced unethical norms of behaviors in quantity measurement processes (Table 2). These behaviors varied across geographies and basically depended on the measuring instruments' nature. In the case of this study, the buyer was using large size measures of the same kind, which is not ethically accepted in the market structure. To mention, 35.5 and 49.5 percent of samples in Dendi and Bako, respectively, encountered this problem. In some aspects, enlarging the size of socially accepted local units was applied (Table 2). Some farmers who used mechanical balance in the marketplace faced another kind of unethical act. According to 15.3 percent of sample respondents, for instance, buyers were using various shrinking crop weight mechanisms. This result implied that mechanical balance itself was not a guarantee to ensure the reliability of measurement. In the two remaining marketplaces, Adea and Gimbichu, acts against the norm were somewhat different. In these sites, different sack sizes were used. The buyers also offered non-equivalent value by undermining units' sizes and amounts (Table 2).

The study concluded that manipulation mechanisms related to bowl and glass units were mostly practiced by buyers by using large sizes of alike units of measurement. In addition, buyers enlarged the size of informally defined measures. Overall, the cereal commodity exchange process entails uncertainty in light of the opportunistic behavior for one of the parties (the farmer) in the exchange process. Theoretically, the productivity of an economy is the function of both technology and institutions of that economy [3]. However, institutions have been given less consideration in the new growth of economics literature. This study, therefore, suggested metrological standardization either of the international measurement system or idiosyncratic national standards. Further, managing of cereals transactional issues through institutions (regulations) have a paramount contribution for controlling unethical behaviors of market actors.

The trading parties' perceptions, correct or incorrect, are the bases of behavior [3]. Most farmers responded that immoral practices of buyers had influences on upon quantity measurement accuracy (Table 2). Measuring instruments-related unethical behavior influences were observed as higher in most study areas (Table 2). Inversely, the influence was perceived lower in the Adea marketplace. This implies that the farmers were losers in cereal commodity exchange due to the existing violence towards informal constraints. To this effect, institutions for cereals quantity measurement processes can maintain the existing norms of behavior and minimize associated measurement costs. Besides, farmers' perception of the accuracy of measuring instruments was analyzed from the closeness of true (ideal) and actual value perspectives. Farmers' true value was seen from the expected value of each local unit. Table 2 results indicate that most farmers agreed on the accuracy of the measuring instrument in all study areas, except for weight balance and can units at Bako Tibe and Adea, respectively. Nevertheless, the percentage ratio of farmers' perception explicitly indicated some sort of unit accuracy problems.

**Table 2.** The farmers' perception of buyers' measurements-related unethical behavior and local unit accuracy.

| Items | | Research Site | | | |
|---|---|---|---|---|---|
| | | Dendi (Count %) | Bako (Count %) | Adea (Count %) | Gimbichu (Count %) |
| Buyer's unethical behavior toward quantity measurements | Yes | 97 (78.2) | 66 (72.5) | 55 (61.1) | 49 (63.6) |
| | No | 27 (21.8) | 25 (27.5) | 35 (38.9) | 28 (36.4) |
| Unethical behaviors types | Expanding the size of measuring instrument | 30 (24.2) | 0 (0) | 12 (13.3) | 10 (13) |
| | Using large size instruments similar to one accepted for exchanging | 44 (35.5) | 45 (49.45) | 0 (0) | 0 (0) |
| | Count deception | 4 (3.2) | 8 (8.79) | 0 (0) | 0 (0) |
| | Measuring below the standard of measuring instruments (shrinking) | 19 (15.3) | 13 (14.28) | 0 (0) | 0 (0) |
| | Offering non-equivalent value for farmers by undermining units' size and amount | 0 (0) | 0 (0) | 43 (47.8) | 39 (50.6) |
| | Did not face at all | 27 (21.8) | 25 (27.5) | 35 (38.9) | 28 (36.4) |
| Buyers' immoral behavior influence on local units' accuracy | Yes | 124 (100) | 91 (100) | 76 (84.45) | 76 (98.7) |
| | No | 0 (0) | 0 (0) | 1 (1.11) | 1 (1.3) |
| | Don not know | 0 (0) | 0 (0) | 13 (14.44) | 0 (0) |
| Farmers' perception of buyers' immoral behavior influence extent upon the local units' accuracy | Very High | 14 (11.3) | 22 (24.2) | 10 (11.1) | 3 (3.9) |
| | High | 52 (41.9) | 30 (32.9) | 34 (37.8) | 20 (26) |
| | Medium | 19 (15.3) | 18 (19.8) | 17 (18.9) | 14 (18.2) |
| | Low | 27 (21.8) | 15 (16.5) | 25 (27.8) | 19 (24.7) |
| | Very Low | 12 (9.7) | 6 (6.6) | 4 (4.4) | 21 (27.3) |
| Farmers' perception of bowl accuracy | Agree | 54 (58.1) | | | |
| | Disagree | 39 (41.9) | | | |
| Farmers' perception of balance accuracy | Agree | 25 (80.6) | 5 (38.5) | | |
| | Disagree | 6 (19.4) | 8 (61.5) | | |
| Farmers' perception of glass accuracy | Agree | | 44 (56.4) | | |
| | Disagree | | 34 (43.6) | | |
| Farmers' perception of sack accuracy | Agree | | | 52 (72.2) | |
| | Disagree | | | 20 (27.8) | |
| Farmers' perception of can accuracy | Agree | | | 4 (22.2) | 7 (63.6) |
| | Disagree | | | 14 (77.8) | 4 (36.4) |
| Farmers' perception of sack accuracy | Agree | | | | 48 (72.7) |
| | Disagree | | | | 18 (27.3) |
| Total | | 124 (100) | 91 (100) | 90 (100) | 77 (100) |

Source: Field survey, 2017.

### 3.4. Association between Local Units' Accuracy and Buyers' Unethical Behavior

The $\chi^2$ test was used to examine the relationship between local units' accuracy and measuring instrument-related buyers' immoral behavior (farmers' perception). The statistical results showed that local units' measurements accuracy and measuring instruments related buyers' immoral behavior were dependent on each other for a bowl, glass; sack and can; and sack units in Dendi, Bako, Adea, and Gimbichu districts, respectively (Table S1). Accordingly, the $\chi^2$ test indicated that the farmers' perceptions of local units' measurements accuracy and measuring instruments related buyers' dissipating behavior had a significant relationship when bowl, glass, sack, and can local units were employed for cereals trade (Table 3).

**Table 3.** Chi-square tests.

| Research Site | Local Units | | Value | df | Asymptotic Significance (2-Sided) |
|---|---|---|---|---|---|
| Dendi | Bowl | Pearson Chi-Square | 31.257 | 4 | 0.000 *** |
| | | Likelihood Ratio | 39.950 | 4 | 0.000 |
| | | Linear-by-Linear Association | 22.253 | 1 | 0.000 |
| | | *n* of Valid Cases | 93 | | |
| | Weight balance | Pearson Chi-Square | 5.800 | 4 | 0.215 |
| | | Likelihood Ratio | 7.834 | 4 | 0.098 |
| | | Linear-by-Linear Association | 4.521 | 1 | 0.033 |
| | | *n* of Valid Cases | 31 | | |
| Bako-Tibe | Glass | Pearson Chi-Square | 17.657 | 4 | 0.001 *** |
| | | Likelihood Ratio | 20.403 | 4 | 0.000 |
| | | Linear-by-Linear Association | 17.315 | 1 | 0.000 |
| | | *n* of Valid Cases | 78 | | |
| | Weight balance | Pearson Chi-Square | 5.113 | 3 | 0.164 |
| | | Likelihood Ratio | 6.774 | 3 | 0.079 |
| | | Linear-by-Linear Association | 2.594 | 1 | 0.107 |
| | | *n* of Valid Cases | 13 | | |
| Adea | Sack | Pearson Chi-Square | 35.361 | 4 | 0.000 *** |
| | | Likelihood Ratio | 42.913 | 4 | 0.000 |
| | | Linear-by-Linear Association | 28.643 | 1 | 0.000 |
| | | *n* of Valid Cases | 72 | | |
| | Can | Pearson Chi-Square | 10.286 | 3 | 0.016 ** |
| | | Likelihood Ratio | 11.431 | 3 | 0.010 |
| | | Linear-by-Linear Association | 7.633 | 1 | 0.006 |
| | | *n* of Valid Cases | 18 | | |
| Gimbichu | Sack | Pearson Chi-Square | 17.306 | 4 | 0.002 ** |
| | | Likelihood Ratio | 21.903 | 4 | 0.000 |
| | | Linear-by-Linear Association | 6.903 | 1 | 0.009 |
| | | *n* of Valid Cases | 66 | | |
| | Can | Pearson Chi-Square | 4.662 | 3 | 0.198 |
| | | Likelihood Ratio | 5.597 | 3 | 0.133 |
| | | Linear-by-Linear Association | 4.063 | 1 | 0.044 |
| | | *n* of Valid Cases | 11 | | |

Where *** and ** are significant *p*-value at 1 and 5 percent, respectively.

On the other hand, $\chi^2$ values for weight balance unit of measurement in Dendi and Bako districts, and can unit in Gimbichu were insignificant at $p < 0.05$. This finding revealed that the two variables had no significant association for weight balance and cans. In general, the result implied that most of the local units related trading partners' abuses behaviors significantly affected measurements accuracy except for weight balance and cans in the Gimbichu study area.

## 4. Conclusions

Lately, most people in Ethiopia rely on local marketplace organizations for lots of their agricultural output transactions and food sources. In this case, the reliability of the measurement system is unquestionably fundamental to ensure equitable economic exchanges. However, this study found that farmers were using multiple and non-uniform local units for carrying out cereals transactions. In addition, the dissimilarity of local units of similar type, method of measuring, and using instruments were enormously observed within the marketplace of studied districts. The findings illustrate that the conversion convention of local units' measurements to metric units was different. There was a difference between home and marketplace-based cereals quantity measurements. Comparatively, the home-based cereals quantity measurements per sack were higher than measurements made in the marketplace.

Moreover, the measurement behavior concerning ownership over the local units and related decisions were highly confined and complicated. In this case, negotiation conflicts in the course of instrument choice, among alternatives, just to execute cereals measurements, was relatively higher when buyers' ownership and decisions were dominant. Furthermore, the norms of offering cereals gifts for the buyers was another component of the local measurement systems. The two-hand palm gift offered for two buyers on average was logically regarded as costs, as the expected social capital outcomes, as a result of the gifts, were almost inconsequential and uncommon across the districts and local units. On the other hand, the violence of buyers towards informal norms of behavior and social conventions of local marketplace's measuring instruments were vastly practical.

In this study, using large size units which were not accepted by marketplace informal norms, shrinking weights, and offering non-equivalent value by undermining units' size and amount were dominantly verified. The findings additionally pinpointed that standard weight balance was not adequately alleviating transactional problems and measurements unreliability. Hence, the assumption of ensuring measurement reliability using standardized measures was not functional to the contexts. This fact also shows that Barzel's measurement error approach was only identifying a part of the measurements problem causes. The measurement-related issues at the transactional level in the local agricultural marketplace, therefore, should be viewed from the broader Velker's sameness paradigm.

In general, the study concluded that the farmers' cereals quantity measurement behaviors, the multiplicity and non-uniformity of local measures, the anomalies of conversion conventions of local units, ownership and choice decisions of appropriate units, the two-hand palm cereals gift, the different home and marketplace quantity measurements, and the buyers' unethical conducts proved the existence of measurement unreliability. The local marketplace unreliability of measurement systems, in turn, resulted in transactions, measurement, social capital, and the two-hand palm cereal gift costs. Thus, the study signified that complete standardization of cereals quantity measurement systems and institutional trade aid are jointly essential to augment the local marketplace measurement matters. To these ends, this study is vital in the context of developing countries and/or nations as a whole for two major aspects. Firstly, the study paves a way to minimize transaction and/or measurement costs by ensuring local marketplace measurement reliability and equitable exchange of products. Secondly, this study is crucial to establish market institutions in this context. In general, the findings of this study have paramount inference for the improvement of agricultural products trade performances, macroeconomic policy planning, national markets integration, and rural incomes, particularly, in developing countries where there are critical problems to standardize units of measurement.

**Supplementary Materials:** The following are available online at http://www.mdpi.com/2077-0472/8/12/188/s1, Table S1: The cross tabulation results between farmers' perception of local units' accuracy and buyers' units-related immoral behavior influence extent.

**Author Contributions:** Conceptualization, K.A.A. and D.Z.; Data curation, K.A.A.; Formal analysis, K.A.A. and B.G.E., F.M.S.; Investigation, K.A.A.; Methodology, K.A.A.; Project administration, D.Z.; Supervision, D.Z.; Validation, B.G.E., D.Z., and R.O.; Writing original draft, K.A.A.; Writing review & editing, B.G.E.,R.O., and D.K.D.

**Funding:** This research was funded by the Fundamental Research Funds for the Central Universities, Grant Number 2662017PY071 and The APC was funded by Deyi Zhou.

**Acknowledgments:** The authors wish to thanks Deyi Zhou for his valuable inputs at all phases of the research. He has a vital contribution in developing and pinpointing areas and gaps in the literature and providing guidance throughout the research endeavor. The authors also thanks Bekele Gebisa for his assistance in reviewing and consulting during the manuscript preparation.

**Conflicts of Interest:** The authors declare no conflict of interest.

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
