# Peer review of "Cereal Commodity Trading in Ethiopian Local Marketplace: Examining Farmers’ Quantity Measurement Behaviors"

_agriculture, doi:10.3390/agriculture8120188_

Round 1

Reviewer 1 Report

The paper is interesting as it deals with the problem of inconsistency of measurement practices across the country. Although it highlights the need to conduct this study, it is not clear to the reader as to why there is a need to have a common measure (needs to strengthen the significance of the study) as well as the policy implications of the research. Further, a theoretical framework linking the heterogeneity of measurement and the perceived immoral behaviour of buyers is needed to help the authors formulate their hypotheses and designed their method of analysis. As it is now, the sampling procedure is not discussed nor the results fully explained.

Author Response

See attached coverletter.

Reviewer 2 Report

This paper should probably be re-sent to this journal after passing a thorough language revision. The text is full of basic mistakes that make any reasonings very hard to follow. This is a huge problem indeed.

Apart from that, the paper is concerned with the different measurement units that are used in cereal trade in Ethiopia. It is argued that the utilization of multiple, non-uniform and non-standardized measurement units causes widespread acts of abuse, injustice and violence. The authors perform some simple statistical exercises, linking the farmers’ perception of the inaccuracy of the measures with the immoral behaviour of buyers (also farmer’s perceptions).

As mentioned before, I may have missed some points due to the language barrier. However, I still think that the authors must convey much more information. They should try to frame the Ethiopian case-study within the broad economic literature on institutions (at this moment, references to the work of Douglas North and others seem a little bit out of place). Perhaps the authors should start the paper with a brief state of the art regarding the importance of homogeneity in measurement units, framing this review within the much broader institutionalist approach to Economics. Then they should try to describe the specific mechanisms through which the heterogeneity in measurement units may translate into acts of abuse or violence (drawing on their experience with Ethiopian farmers). This would probably help them to draw more comprehensive conclusions from the statistical analysis.

Author Response

See attached coverletter.

Reviewer 3 Report

First of all, I would like to congratulate the authors for the paper, which is very interesting and whose theme is totally related to the Agriculture journal.

Regarding formal aspects, the structure of the article is perfect; however, the bibliographical references are scarce. I believe that a more extensive revision should be made. Moreover, the authors have to review the way to quote and some spelling errors. 

Regarding Material and Methods, the description of study area is good, although it would be appropriate for the authors to add a geographical characterization of the study area in order that the readers understand the activity on cereal production.

In the Sampling Methods section, the author should add the formula used to determine the sample. Finally, reading the section, I really don’t know what methodology of analysis has been used because it is not explained. Furthermore, a list of variables used in Table 1 should be explained as well as the source.

Author Response

See attached coverletter.

Round 2

Reviewer 2 Report

The authors have made major changes and have significantly improved the manuscript. 

Reviewer 3 Report

With the changes introduced this paper  can be published